# Utilization of Agro-Industrial Residues in the Rearing and Nutritional Enrichment of *Zophobas atratus* Larvae: New Food Raw Materials

**DOI:** 10.3390/molecules27206963

**Published:** 2022-10-17

**Authors:** Renata Quartieri Nascimento, Cláudio Vaz Di Mambro Ribeiro, Nelson Barros Colauto, Larissa da Silva, Paulo Vitor França Lemos, Ederlan de Souza Ferreira, Giani Andrea Linde, Bruna Aparecida Souza Machado, Pedro Paulo Lordelo Guimarães Tavares, Aline Camarão Telles Biasoto, Marcelo Andres Umsza Guez, Natália Carvalho, Denílson de Jesus Assis, Jania Betânia Alves da Silva, Carolina Oliveira de Souza

**Affiliations:** 1Graduate Program in Biotechnology/RENORBIO, Institute of Health Sciences, Federal University of Bahia, Salvador 40100-000, Brazil; 2Graduate Program in Food Science, Faculty of Pharmacy, Federal University of Bahia, Salvador 40170-115, Brazil; 3Department of Animal Science, School of Veterinary Medicine and Animal Sciences, Federal University of Bahia, Salvador 40170-110, Brazil; 4Faculty of Pharmacy, Federal University of Bahia, Salvador 40170-115, Brazil; 5SENAI Institute of Innovation (ISI) in Health Advanced Systems (CIMATEC ISI SAS), University Center SENAI/CIMATEC, Salvador 41650-010, Brazil; 6Brazilian Agricultural Research Corporation (EMBRAPA), Jaguariúna 13820-000, Brazil; 7Department of Biotechnology, Institute of Health Sciences, Federal University of Bahia, Salvador 40110-100, Brazil; 8Graduate Program in Chemical Engineering, Polytechnic School, Federal University of Bahia, Salvador 40210-630, Brazil; 9School of Exact and Technological Sciences, Salvador University, Salvador 40210-630, Brazil; 10Mechanical Engineering Collegiate, Center of Science and Technology, Federal University of Recôncavo of Bahia, Cruz das Almas 41770-235, Brazil

**Keywords:** *Zophobas morio*, food conversion, by-product, insect, sustainability

## Abstract

Edible insects are a potential alternative food source of high feed conversion efficiency and protein content. *Zophobas atratus* is an edible insect that adapts to different diets, enabling sustainable rearing by adding value to by-products and agro-industrial residues. This study aimed to evaluate the performance and nutritional characterization of *Zophobas atratus* larvae fed with different proportions of grape residue. Physicochemical analysis of the diets and larvae (AOAC procedures), fatty acid profile (chromatographic techniques), metals and non-metals (inductively coupled plasma optical emission spectrometry), larval mass gain, feed conversion efficiency, and mortality rate were assessed. The replacement of 25% of the conventional diet with grape residue increased lipid, ash, and fiber contents and reduced protein, carbohydrates, and energy. It promoted greater mass gain, lower mortality rate, and reduced larval growth time by 51%. Among the replacements, 25% resulted in the second-highest content of calcium, sodium, magnesium, and zinc, and the lowest content of potassium and phosphorus in the larvae. The 100% replacement resulted in the highest amounts of C18:2n6 (27.8%), C18:3n3 (2.2%), and PUFA (30.0%). Replacing 25% of the conventional diet with grape residue is equivalent to the conventional diet in many aspects and improves several larvae performance indices and nutritional values.

## 1. Introduction

Edible insects have been proposed as an alternative food to replace conventional animal-based proteins [1], and although unfamiliar to consumers, entomophagy has been growing worldwide [2]. This trend is a way to meet an estimated demand from the world population of 9.7 billion people in 2050. The production and supply of animal protein obtained by conventional methods will not be sufficient to meet the high demand [3,4]. This scenario implies a potential lack of food security in an environment with increasingly scarce and limited resources [5]. Insects could be a possible solution to supply food for a growing population [1,6,7].

Insect farming has a lower environmental impact and better use of natural resources than raising conventional animals such as cattle, pigs, and poultry [8,9]. Therefore, allowing feasible, sustainable, and innovative food [10,11]. The insects can have up to 30% more protein content than conventional sources [12]. They present expressive amounts of essential amino acids such as leucine, lysine, and methionine, dietary fiber, polyunsaturated fatty acids such as omega 3, 6, and 9, micronutrients [13], and vitamins A, C, E, and B complex [14,15]. The nutritional values vary depending on the species, rearing conditions, environment, diet, and stage of development [16,17,18,19]. However, the negative perceptions of insects as food undermine the global market, and the lack of systematic work, legislation, and regulations to assure safety, shelf life, standardization, and quality control of insect rearing and trade is still an obstacle in the edible insect industry [2].

The best-known and reared edible insects (*Coleoptera*: *Tenebrionidae*) in the world are the yellow mealworm *(Tenebrio molitor* L.), lesser mealworm beetle *(Alphitobius diaperinus* Panzer), and giant mealworm *(Zophobas atratus* Fabricius) [20]. The European Food Safety Commission established by regulation 2015/2283 the use of *T. molitor* larvae for human consumption. Cereal bars, pasta, cookies, and animal feed can be directly prepared or added by dry insect biomass [21], providing further clarification and confidence for the safe use of insects as a food source.

*Zophobas atratus* is an edible insect with great nutritional value, digestibility, and feed conversion rate. This insect has a high ability to adapt to different production systems and food sources from agro-industrial residues, making its rearing more sustainable [22,23]. The use of agro-industrial by-products in insect feeding reduces economic pressure and the demand for soy, wheat, and corn-based feeds used for other animals [24]. Therefore, decreasing the competition for these commodities that are also used in human nutrition [20].

Most studies involving agro-industrial residues for feeding insects focus on *T. molitor,* and according to the diet provided, this insect has shown great variation in feed conversion, growth performance, and nutritional profile [25]. Other edible insects such as *A. diaperinus* and *Z. atratus*, less studied, may also present variable performance depending on the diet used. Beet molasses, potato skins, fermented brewery grains, bread and cookie leftovers, wheat bran, and olive processing residues are the agro-industrial by-products most used to formulate diets for edible insect larvae [24,26,27,28]. The growth conditions for each larval instar of *Z. atratus* have been studied regarding temperature, humidity, photoperiod, and diet. The diet is one of the main sources of variation in larval growth [29]. Although several agro-industrial by-products have already been reported, no studies have been found using grape residues to feed insects. Therefore, this study aimed to evaluate the performance and nutritional characterization of *Z. atratus* larvae fed with different proportions of grape residue (25, 50, 75, and 100%) in the conventional diet (control).

## 2. Results and Discussion

### 2.1. Nutritional Composition of Diets

Diet D1 had 2.8 times more protein, 1.3 times more carbohydrates, and 1.2 times more energy than diet D5 (Table 1). On the other hand, D5 had 3.2 times more lipids, 1.1 times more ash, and 6.2 times more fiber than D1 (Table 1). Overall, the increase in grape residue in diets from D1 to D5 resulted in a proportional increase in lipids, ash, and fiber values and a proportional reduction in protein, carbohydrate, and energy (Table 1).

### 2.2. Larvae Growth Performance

The cumulative mass gain of larvae increased over time for all treatments and controls. However, there was a greater mass gain for D1 and D2, and a smaller mass gain for D3, D4, and D5 (Figure 1). This evidences that the greater addition of grape residue in the diet implies a smaller proportional mass gain of the larvae. However, the mass gain in D1 (control) and D2 were equal (*p* > 0.05), indicating that the replacement of 25% of the conventional diet with grape residue did not result in differences in the mass gain of the larvae and that both were generally better (*p* ≤ 0.05) than the other treatments.

The cumulative mass gain of larvae in 44 days reached 98% of the total mass in 90 days of rearing for D1 (52.5 g), 91% for D2 (46.8 g), 79% for D3 (36.6 g), 76% for D4 (28.33 g), and D5 (17.0 g). Thus, the rearing time was about 50% shorter for D1 and D2 to obtain more than 90% of the expected larval mass for 90 days of growth. There was no difference (*p* > 0.05) in the cumulative mass gain of larvae during rearing between D1 and D2; although, D1 presented nominal values greater than D2 between 30 and 60 days (Figure 1). At 90 days of rearing, the cumulative mass gain was higher (*p* ≤ 0.05) for D1, D2, and D3, followed by D4 and then D5 (Figure 1). Therefore, the D4 and D5 diets are inadequate for the larvae to reach the maximum growth potential compared to the control and other treatments.

Larvae of *T. molitor* showed greater daily mass gain when fed with brewery grains containing 18% protein than larvae fed with 7% protein in the diets [26]. In the present study, the greater mass gain of *Z. atratus* larvae may have been affected by the greater protein availability in the D2 and control diets.

The replacement of 25% of the standard diet (100% wheat) with olive residue in the rearing of *T. molitor* was not adequate for mass gain and was justified by the better balance and availability of carbohydrates and proteins and lower levels of lipids [28], similar to the contents of the diets of the present study. The reduction in the mass gain of larvae fed with a greater amount of grape residue in the diets (D3, D4, and D5) in the present study may have been due to the selective feeding preference of insects, in general, for proteins and lipids [20,30,31].

The daily mass gain of larvae as a function of rearing time was, in general, consistently higher (*p* ≤ 0.05) for D1 and D2 in contrast to D3 (32% lower), D4 (52% lower), and D5 (66% lower), than the control at 30 days of growth (Table 2). Therefore, replacing the conventional diet with up to 25% of grape residue resulted in equal values of larvae mass gain. The larvae mass gain values from 31 to 60 and 61 to 90 days of breeding were 89% on average lower than the control from 0 to 30 days. Therefore, the first 30 days of growth were more effective than the 31 to 90 days.

The greater daily mass gain of the larvae, mainly in the first 30 days of rearing, may be related to the greater demand for nutrients in the initial stages of growth. In turn, the lower mass gain in the adult larval stage may be associated with the senescence of this stage. The exhaustion of nutrients over time caused by the larvae and the accumulation of waste at the place of rearing contaminate the nutritional source, making it toxic. Thus, favoring the growth of competing and antagonistic microorganisms.

Explanations for differences in mass gains during the rearing periods corroborate with other reports. Li et al. [31] reared *T. molitor* larvae fed on wheat residues and fermented vegetables and reported that after 15 days, there was a reduction in the nitrogen content of the diet, which would be necessary to maintain the larvae’s mass gain. The authors’ analysis suggests that the periodic exchange of food supply or protein replacement supports larval growth.

The ECI was higher (*p* ≤ 0.05) for D1 and proportionally reduced from D2 to D5 (Figure 2—ECI). ECI indicates how much of the ingested food has been transformed into animal mass. The control diet was the most efficient, followed by D2 because the higher the ECI value, the greater the breeding efficiency.

The FCR was lower (*p* ≤ 0.05) for D1 and proportionally increased from D2 to D5 (Figure 1—FCR). FCR demonstrates the amount of food that needs to be ingested by the animal to obtain one kilogram of mass. Therefore, the lower the FCR value, the more efficient the rearing.

The MR of larvae was lower in treatment D2, followed by D1 (control) and other treatments (Figure 1—MR). This evidences that the addition of 25% of the grape residue in the conventional diet provided 60% more survival of the larvae concerning the control. Probably, D2 must contain favorable compounds in sufficient quantity for the growth of larvae capable of controlling natural competitors, including antagonistic microorganisms, promoting a more favorable breeding environment.

Larva-fed diets with higher amounts of protein (D1 and D2) had the highest ECI values. Therefore, indicating that the greater availability of protein in the diet promotes greater feed conversion efficiency of the larvae. Conversely, the lowest protein availability in the diets with the highest grape residue (D4 and D5) showed the lowest ECI values. Oonincx et al. [24] evaluated diets with different proportions of protein and lipids and found that dietary protein content is a primary determinant of ECI efficiency. Insect feeding is the main variable in determining the efficiency of the ECI, according to de Vries and de Boer [32] and Wilkinson [33].

The ECI values of larvae fed on diets D2 (17.8%) and D3 (17.3%) were higher (15.8%) than those reported by van Broekhoven et al. [20] for *Z. atratus* larvae fed with diets (organic by-products) containing low protein contents (11.9%), but lower (28.9%) than those fed with high protein diets (32.7%). Zhang et al. [25] reported higher ECI values (36–56%) for *Z. morio* fed with residues; however, the diets had much higher protein values (43.2%) than the diets used in our study (3.9–9.0%). In conventional rearing animals, the energy content of the feed determines growth rates and efficiencies [34]. On the other hand, in insects, protein density and composition are more important [24,35], mainly because they do not use energy to maintain constant body temperature. Moreover, protein-rich diets resulted in lower FCR and higher ECI for most species. Applying the ECI-based calculation, *Z. atratus* larvae are estimated to increase 1 kg in body mass when 4.15 kg (D1), 5.62 kg (D2), 5.80 kg (D3), 6.80 kg (D4), or 10.67 kg (D5) of the diet was provided in our study. The high requirement shown by diets D4 and D5 is probably due to their low nutrition, containing high fiber contents (12.2 and 15.5%, respectively) and low protein contents (5.6 and 3.9%, respectively). However, it is worth noting that *Z. atratus* rearing, even on a diet containing 100% grape residue (D5), was more efficient than bovine meat production (25 kg of feed per 1 kg of animal mass) and diets D2, D3, and D4 were more efficient than pig and cattle production (9.1 kg and 25 kg of feed per 1 kg of animal mass, respectively) [36].

The FCR for the larvae was 3.76 g/g for D2, the best result among the treatments; however, it was higher than the control (3.19 g/g). The FCR of both D2 and control treatments were higher than those reported for carrot-fed *T. molitor* larvae—FCR of 2.2 g/g [9,24]. These results indicate that there may be a possibility of decreasing FCR values, increasing the conversion efficiency for *Z. atratus* larvae. The FCR values determined in the present study for *Z. atratus* larvae are similar to those of swine production (3.6 g/g) and better than that of cattle production (7.8 g/g) [33]. However, the edible portion of the larvae is about 50% larger than that of birds or swine, which would be another efficiency factor in the insect production process [32].

The energy content of the diet determines growth rates and feed conversion efficiency in conventional animal production [34]. Insects make food conversion more efficient due to not using the energy from the diet to keep their body temperature. The protein content is the most critical diet component for these poikilothermic animals [35].

The higher MR of larvae fed with diets with higher addition of grape residue (D3, D4, and D5) can be explained by the higher content of phenolic compounds from this residue (1031.0 mg/kg; *Vitis* spp. cultivar BRS Magna) [37]. Some phenolic compounds were identified from the winemaking process, such as catechin, epicatechin, and gallic, caffeic, syringic, vanillic, *p*-coumaric, and *o*-coumaric acids in grape (*Vitis vinifera* L.) residues (skins and seeds) [38] that were reported with different levels of toxicity to insects [39]. Larvae of *T. molitor* fed on olive residues suggested that phenolic compounds in the diet could be responsible for the higher mortality rate of insects [28]. Phenolic compounds can form complexes with proteins, affecting the digestibility, absorption, and assimilation of amino acids, resulting in increased larval mortality and reduced growth rate and survival [40,41].

### 2.3. Analysis of the Main Components of Nutritional Composition and Larval Performance

The analysis of the principal components was carried out from the bromatological analysis of the diets. The first two components (PC1 and PC2) explained 94.0% of the data variation (Figure 3). PC 1 had most of its variation due to the contrast of treatments, while PC2 was best represented by the variation in the control treatment (D1). Three clusters were found from the results of the bromatological analysis of the diets, in which the protein content, water activity, and moisture form a cluster inversely correlated with the cluster composed of lipids and fibers (Figure 3). The third grouping consisted of carbohydrates and energy (Figure 3).

Larvae fed with diets (D1—control, D2, and D3) that contained the lowest concentration of grape residue were plotted on the right side (positive and negative) together with protein, water activity, moisture, and mass gain of the larvae (Figure 3). This relationship evidences that diets with higher protein, water activity, and moisture values favor the performance of larval mass gain and the increase in the content of these parameters. On the other hand, larvae fed with diets with higher grape residues (D4 and D5) were positioned in the left portion of the graph (positive and negative), together with the variables lipids and fiber, being in opposition to the larvae’s mass gain performance (Figure 3). Thus, the reduction in larval mass gain performance may be related to the diet’s content of lipids and fiber.

Despite the knowledge about the biology of mealworm larvae, little is known about their specific metabolic and nutritional response to dietary conditions [42]. Melis et al. [27] suggest that protein, lipid, and fiber contents in diets may interfere with the metabolism of insects that are able to convert plant fibers by enzymes in their digestive tract. However, Dreassi et al. [43] reported that the total fat content of *T. molitor* was not affected by the fat content in the diet. On the other hand, van Broekhoven et al. [20] reported that mealworm fat content is directly affected by the fat content in the diet and suggest that lipid biosynthesis from carbohydrates may be active in larvae reared on low nutritional quality substrates. These contradictory results are related to the substrate composition variability and may be attributed to the selective larval uptake of fatty acids from the diets and/or modulation of biosynthetic pathways, making an analysis with potential generalizations quite challenging. For protein, van Broekhoven et al. [20] observed no significant differences in the protein content of larvae reared on diets with three levels of protein content. In addition, larvae from the *Tenebrionidae* family have been reported with the ability to regulate their growth metabolism, such as humidity and temperature [44]. Thus, it suggests that the larvae are able to adjust their metabolism and select nutrients from the substrate to achieve nutritional balance for better development [45].

### 2.4. Larvae Physicochemical Characterization

Overall, the replacement of 25 and 50% of the diet with grape residue resulted in higher protein values (*p* ≤ 0.05) in the larvae, and the 25% replacement resulted in higher values of lipids and energy (*p* ≤ 0.05), compared to control and other treatments (Table 3). However, the larvae’s carbohydrate, ash, and fiber values were higher (*p* ≤ 0.05) in the treatments with the higher addition of grape residue in the diet. The water activities of D3 and control were higher than the other treatments (*p* ≤ 0.05) (Table 3).

The ash contents of the larvae ranged from 3.0 to 3.9 g/100 g. Larvae fed with D1, D2, and D3 had the lowest ash contents (*p* ≤ 0.05), which increased in a directly proportional way to the grape residue added in each of the treatments (Table 3). According to Makkar et al. [46], the levels of metals and non-metals in insects of the *Tenebrionidae* family are less than 5%, which corroborates the values found in the present study for *Z. atratus* larvae. However, the diet with 100% grape residue resulted in a 30% increase in ash content compared to the control. This increase indicates an effect of diet on the metal and non-metal composition of larvae. According to Klasing et al. [47] and Payne et al. [17], edible insects are sources of calcium, iron, zinc, and potassium and their contents are highly variable and affected by diet composition.

The carbohydrate contents of the larvae ranged from 2.4 to 7.4 g/100 g. Larvae fed with D1 and D2 had the lowest levels of carbohydrates (*p* ≤ 0.05), which increased in a directly proportional way to the grape residue added in each of the treatments (Table 3). According to Mancini et al. [26], the higher larvae carbohydrate values are related to the lower lipid content and higher moisture content in the diets, similar to the results obtained in the present study.

The fiber content in the larvae ranged from 5.7 to 8.6 g/100 g. Larvae fed with D3 and D5 had the lowest and highest fiber contents (*p* ≤ 0.05), respectively, but without a proportional relationship with the addition of grape residue in the diets (Table 3). Finke [48] reported the highest amount of chitin in the exoskeleton of adult insects and that fiber content could be directly affected by diet. Diets with a higher fiber content are associated with reduced cardiovascular disease, glycemic index, and obesity [49]. Fibers in insects are largely found in the exoskeleton as chitin. The larvae of the present study showed higher fiber content with a diet with 100% grape residue, which is a factor that can improve the quality and welfare of the insect.

The water activity of the freeze-dried larvae ranged from 0.046 to 0.063. Larvae fed with D3 and the control diet had the highest water activities (*p* ≤ 0.05), which decreased in a directly proportional way to the grape residue added in each of the treatments, except for D2 (Table 3). The water activity in food is essential to ensure its storage quality, and values greater than 0.9 favor the proliferation of microorganisms, while values below 0.6 inhibit microbial growth [50]. After freeze-drying, the water activities were ≤ 0.6 for the control and D1–D5, meaning storage-safe diets. However, depending on the drying process used, it is recommended to carry out microbiological tests to ensure storage quality.

The protein contents of the larvae ranged from 39.6 to 42.6 g/100 g. Larvae fed with D2 and D3 had the highest protein contents (*p* ≤ 0.05), followed by D4, control, and D5, indicating that adding 25 and 50% of grape residue in the diet increased the protein content in the larvae compared to the control (Table 3). Protein levels vary from 40 to 52% for *Z. atratus*, from 58 to 67% for *A. diaperinus*, and from 45 to 69% for *T. molitor* [18,25,51,52,53]. According to Gao et al. [54], *T. molitor* larvae showed protein level variations due to their ability to select the nutrients they ingest. However, when the diet offers no choice because it is uniform, it can affect the larvae’s protein content and performance.

The protein levels found in the present study for *Z. atratus* larvae demonstrate that they can convert waste into a valuable product. The obtained biomass can be exploited for human and animal feed as an alternative protein source to reduce production costs [1,28]. It is noteworthy that the protein levels reported in this study are equivalent to conventional protein sources such as chicken (42.6%) and soy (38.2%) [55].

The larvae presented total lipid contents between 40.5 and 45.6 g/100 g. Larvae fed with D2 had the highest lipid content compared to the control (*p* ≤ 0.05), which decreased in a directly proportional way to the grape residue added in each of the treatments (Table 3). Previously reported lipids levels in insects ranged from 21 to 45% [19,28,56] and are compatible with those obtained in the present study. The lipid levels in the larvae are also directly influenced by the composition of the diet [20,24]. According to Arrese and Soulages [57], the lipid content in the larvae can be negatively affected when there is a low supply of lipids in the diet, forcing them to use their lipid reserves. However, in the present study, the opposite occurred. The greater supply of lipids in the diet resulted in lower lipid content in the larvae and vice versa. *Z. atratus* larvae converted grape residue and its combinations into protein and lipid biomass, a valuable product due to its nutritional value [20,25].

### 2.5. Comparisons of Larvae Nutritional Properties

The average content of treatments and control for larvae protein was 41.4 g/100 g, equivalent to 97% of a chicken drumstick, 108% of soybean meal, and 74% of cod (Table 3) [55]. The highest protein values (*p* ≤ 0.05) in larvae were found in D2 and D3 (42.6 and 42.3 g/100 g, respectively) compared to control and other treatments (Table 3). The protein values of D2-fed larvae represent 67.8, 77.7, and 77.9% of the protein values for pork shank, egg, and strip loin beef, respectively (Table 3) [55].

The average content of treatments and control for larvae’s lipids was 43.4 g/100 g, a value equivalent to 102% of that present in strip loin beef, 119% of egg, and 208% of soybeans (Table 3) [55]. The lipid values of the larvae fed with D2 (45.6%) represent 294, 145, and 135% of the lipid values for soybean meal, salmon, and pork shank, respectively (Table 3) [55].

Even the lowest protein values (39.6%; *p* ≤ 0.05) of the larvae fed with D5 already represent 92, 65, and 70% of the protein values of the chicken thigh, pork shank, and fish (cod), respectively. Similarly, the lowest lipid values in D5 (4.5%; *p* ≤ 0.05) represent 261, 95, and 111% of the lipid values of soy flour, strip loin beef, and egg, respectively (Table 3) [55]. Thus, the protein and lipid values of the larvae are comparable to or even higher than those of protein and lipid conventional primary sources of animal and vegetable origins.

The total energy value of the larvae ranged from 552.3 to 590.4 kcal/100 g. Larvae fed with D2 had the highest (*p* ≤ 0.05) energy value compared to the control and were proportionally reduced as the amount of grape residue in the diet of the other treatments increased (Table 3).

Fourteen fatty acids were identified in the larvae of *Z. atratus*. Palmitic (C16:0), oleic (C18:1n-9), and linoleic (C18:2n-6) acids are predominant in all treatments, corroborating with previous studies of species of the same family (*Coleoptera*) [20,25,28,46,51]; although, the contents are widely variable.

The control treatment larvae had the highest amount of palmitic acid (34.4%; *p* ≤ 0.05). This value decreased as the addition of grape residue in the larvae’s diet increased (D2 to D5), indicating that adding residue reduces the amount of palmitic acid in the larvae. The oleic acid content increased (*p* ≤ 0.05) in D2, D3, and control, indicating that amounts above 50% of grape residue in the diet negatively affect the amount of oleic acid in the larvae. Linoleic acid increased (*p* ≤ 0.05) in D4 and D5, indicating that higher amounts of grape residue in the diet favored a higher amount of linoleic acid in the larvae (Table 4).

Insects can be sources of polyunsaturated fatty acids [16,58,59]. The larvae produced in this study presented 2 and 20 times more C18:3n3 and C18:2n6, respectively, compared to strip loin beef, which contains 1.1% of C18:3n3 and 1.8% of C18:2n6 [60]. The highest amounts of fatty acids in insect larvae are affected by diet [16,56], as verified in the present study. However, the higher amount of unsaturated lipids (58.6%—D5) can limit the conservation and processing of insects and their derived products due to oxidation reactions [1].

The larvae of the present study showed a variation of 12 to 14 of the n6/n3 ratio of fatty acids, typical of insect larvae from the *Coleoptera* family [61]. The results obtained were similar to those reported by Siemianowska et al. [62] for *T. molitor* larvae. However, there is a wide range of reported values of the n6/n3 ratio for *T. molitor* from 32 to 44 [43], 16 to 21 [25], and 25 to 63 [28]. The n6/n3 ratio values found in the present study are above those recommended by the WHO, which mentions ten as the reference value [63]; however, it was possible to verify that the adjustment of the diet can change the values of the n6/n3 ratio closer to the recommended.

IA and IT indices are determined to evaluate the risks of fat on human health. The IA was lower in larvae fed with greater addition of grape residue, with values similar to those of *Z. atratus* larvae reported by Mlček et al. [22] of 0.7 and Barroso et al. [64] of 0.6. According to Mlček et al. [22], the value of IA is similar to that of polyunsaturated acids in olive oil. Larvae IT ranged from 1.4 to 1.6 and was higher than the values reported by Finke [51] of 1.3 and by Ramos-Bueno et al. [65] of 1.4 with the same larvae. In general, adding grape residue to the larvae’s diet did not change the IT values.

### 2.6. Composition of Metals and Non-Metals in Larvae

The lower amount of grape residue in the diet promoted a higher concentration of calcium, sodium, magnesium, and zinc in the larvae, mainly in D1. The greater amount of grape residue in the larvae’s diet promoted a greater concentration of potassium and phosphorus in the larvae, mainly in D5 (Table 5). The manganese and iron concentrations of the larvae were the same in all treatments and controls, indicating that these elements are not affected by the addition of grape residue in the diet (Table 5). Other elements such as selenium, arsenic, cadmium, cobalt, copper, and nickel were not found in the larvae considering the detection limits of the method, indicating that there was no inhibition by the presence of these elements.

There is a sense of reluctance and disgust in eating insects by Western consumers, which is associated with primitive behavior in every culture and affects food habits [1]. Thus, insects used as ingredients in processed foods offer great promise to increase acceptance and decrease repulsion by the population [66]. Chocolate cookies replaced with 15% ground cricket were acceptable, as they showed no noticeable change in taste [67]. The use of insects in food is still at an early stage of appreciation and regulation as novel foods, but the increasing authorization of their use in Europe has strengthened the market, facilitating access, with a potential increase in consumption [68] that could be driven by an eventual lower availability of conventional animal protein on the market.

## 3. Materials and Methods

### 3.1. Biological Material

The SuperBugs-*Alimentos Funcionais* (Salvador-BA, Brazil) company donated 15-day-old *Z. atratus* larvae that were fed with a commercial poultry feed (egg-laying chicken; Mais Quintal Crescimento e Engorda; Carbrás Ltd.a, Lauro de Freitas-BA, Brazil) consisting of 18.4% protein, 3.4% lipids, 5.8% fiber, 9.2% ash, 90 mg/kg iron, 10 g/kg calcium, 5.2 g/kg phosphorus, 87 µg/kg selenium, 10.80 mg/kg copper, 108 mg/kg manganese, 108 mg/kg zinc, 2.16 mg/kg iodine, 6.5 g/kg lysine, 3.5 g/kg methionine, 0.03 µg/kg biotin, 0.22 mg/kg thiamine, 5.65 mg/kg vitamin E, 4995 IU/kg vitamin A, 8100 IU/kg vitamin D3, 0.18 mg/kg folic acid, 6.75 µg/kg vitamin B12, 15.75 mg/kg niacin, 2.25 mg/kg riboflavin, 0.45 mg/kg pyridoxine, and 5.62 mg/kg pantothenic acid, without pesticides or antibiotics, as informed by the manufacturer.

### 3.2. Diet Formulation

A conventional diet of 70 g/100 g commercial poultry feed (1.19 mm grain size) and 30 g/100 g commercial ground maize (*Zea mays*; 0.71 mm grain size) grains (Maratá, Itaporanga-SE, Brazil) was used to feed 15-day-old larvae and consisted of 38% carbohydrates, 3.6% proteins, 1.1% lipids, and 2.6% fibers, according to the manufacturer’s information. A conventional diet was used to feed *Zophobas* spp. larvae that were then used to feed fish. This mixture was replaced by zero (control), 25, 50, 75, and 100% (mass/mass) of residue from the production of grape juice (*Vitis* spp. cultivar BRS Magna (a hybrid of cultivars BRS Rúbea x IAC 1398-21 (Traviú) Embrapa) to formulate the diets used in the experiment (Table 6).

This residue originates from the final processing step, composed of 40% skin and 59% grape seed, and 1% impurities supplied by the Brazilian Agricultural Research Corporation (Embrapa) Semiárido (Petrolina, Brazil). The grape residue was subjected to the steam extraction method [69], vertical hydraulic press, and drying in an industrial tray (Ibram VC 3580, São Paulo, Brazil) for 36 h at 40 °C before transport. In the laboratory, the grape residue was ground in a Willye-type knife mill (Tecnal R-TE 650/1, Piracicaba, Brazil) and sieved (25 mesh) to obtain a granulometry smaller than 0.71 mm.

### 3.3. Larvae Cultivation

About 96 larvae averaging 23.47 ± 2.03 g each were reared per plastic box. The cultivation was performed in flat plastic boxes measuring 30 cm × 20 cm × 8 cm (length × height × width), opening at the top (7.5 cm × 15 cm), and covered by a screen (25 mesh). Each box contained an 800 g diet (wet basis) *ad libitum*. Slices of in natura potato (*Solanum tuberosum* L.) with a ratio of 0.36 g/larva were placed in layers of cotton and replaced every two days to preserve moisture. The boxes (n = 3) were randomly placed on shelves in a climatic chamber (TE-4001, Tecnal, Piracicaba, Brazil) at 26 ± 1 °C and 51 ± 4% relative humidity (Minipa MT-241 digital thermo-hygrometer, São Paulo, Brazil) in the dark for 90 days.

Periodically, the larvae were separated from the diet with sieves to measure their mass on an analytical balance, returning to the previous condition without replacing or altering the diet. After 90 days of cultivation, the larvae were separated and transferred to another empty box in the same environment for 24 h [20]. After measuring the larvae mass, they were washed in running water, frozen in a mechanical ultra-freezer at −80 °C for 48 h, and freeze-dried (Lyophilizer L101, Liobras, São Carlos, Brazil) for 72 h. The dried material was ground in a knife mill (Cadence, Piçarras, Brazil), sieved (14 mesh) to obtain particles smaller than 1.4 mm, and stored in glass containers with a nitrogen atmosphere (N_2_) at −80 °C for further analysis.

### 3.4. Physicochemical Analysis

The physicochemical composition of diets and larvae was analyzed according to AOAC [70]. Moisture was determined by drying in an oven (Tecnal, TE-394/I) at 105 °C until constant mass. The ash was determined in a muffle furnace (Lavoisier 402-D, São Paulo, Brazil) at 550 °C and total fiber content by acid and alkaline extraction. The Bligh and Dyer [71] method was used to determine the total lipids. The carbohydrate content was calculated by difference [72]. The energy value was calculated considering 4 kcal/g for carbohydrates and proteins and 9 kcal/g for lipids [73]. Water activity was obtained using Aqualab Lite (AL1612, Decagon Devices, Pullman, WA, USA) at 25 °C. Crude protein content was determined by the Kjeldahl method [70] with defatted larvae [74]. The nitrogen conversion factor was 6.25 for the diets and 4.76 for the larvae [75].

### 3.5. Fatty Acid Profile

Either identification or quantification of fatty acids in the diets and larvae was performed according to Souza et al. [76]. Methylation of total lipids was carried out in organic media containing methanolic NaOH (0.5 M) and boron trifluoride (12 g/100 mL) solutions. The fatty acids methyl esters were extracted with isooctane to proceed with the chromatographic analysis.

The fatty acid methyl esters were separated in a gas chromatograph (Perkin Elmer Clarus 680) with a flame ionization detector and a DB–Fast FAME column (30 m × 0.25 mm × 0.25 μm). The analysis parameters were injector temperature 250 °C, detector temperature 280 °C, oven temperature programmed to 60 °C for 0.5 min, increasing 25 °C/min to 194 °C, remaining at that temperature for 1 min, increasing by 5 °C/min to 235 °C, and remaining at this temperature for 1 min. Helium was used as carrier gas with a 1.0 mL/min flow rate, and 1 μL injections were performed in split mode (1:50).

The identification of fatty acids was by comparing the retention time of the peaks of the samples with the retention time of the fatty acid methyl esters of a standard mixture (C4-C24 189-19, Sigma Aldrich, San Luis, MO, USA). Quantification was performed by normalizing the peak areas. Total saturated fatty acids (SFA), total monounsaturated fatty acids (MUFA), and total polyunsaturated fatty acids (PUFA) ratios were calculated based on the fatty acid profile. The indices of atherogenicity (IA) and thrombogenicity (IT) were calculated [77].

### 3.6. Larvae Growth Performance

Larvae growth performance was calculated according to Zhang et al. (2019). Larvae mass gain was determined by the difference between the initial and final mass of each period of about seven days. The Efficiency of Conversion of Ingested Food (ECI), feed conversion rate (FCR), and mortality rate (MR) of larvae were calculated according to Equations (1)–(3) [20,25]. The average individual consumption was calculated using values of the total consumption of the diet by the average fresh larvae in up to 90 days of rearing.
ECI (%) = (Average mass gain)/(Average individual consumption) × 100 (1)
FCR (g/g) = (Average individual consumption)/(Average mass gain) × 100 (2)
MR (%) = (Dead larvae)/(Initial larvae) × 100 (3)

### 3.7. Analysis of Metals and Non-Metals

The determination of metals and non-metals in diets and larvae was performed by inductively coupled plasma optical emission spectrometry (ICP OES; Agilent Technologies, model 720 series, Santa Clara, CA, USA). The contents of copper, sodium, manganese, magnesium, selenium, iron, calcium, zinc, potassium, phosphorus, cobalt, arsenic, cadmium, and nickel were determined. Analytical curves for each element were linear over the entire working range and covered sample concentrations. The method’s accuracy was established by analyzing reference material and certified apple leaves (NIST 1515) under the same analysis conditions as the samples.

### 3.8. Statistical Analysis

The experimental design was completely randomized. The analyses were performed in triplicate (n = 3) for each experimental group, and the results were expressed as arithmetic means with standard deviation. Data from diets and larvae were submitted to analysis of variance (ANOVA), and differences were compared by Tukey’s test at a significance level of 5% (*p* ≤ 0.05) using the Statistica 8.0 software. Larvae performance was evaluated by non-linear regression using the SAS University Edition 1.7.0_76 Program. Principal component analysis (PCA) was performed using XLSTAT software (Version 2019, Addinsoft, New York, NY, USA) 17.04 to evaluate larvae performance and nutritional characterization of diets.

## 4. Conclusions

The replacement of 25% of the conventional diet with grape residue showed similar results to the control and was more effective than the replacements of 50, 75, and 100%. Moreover, 25% grape residue in the conventional diet promoted expressive mass gain, lower mortality rate, and reduced larval rearing time. Moreover, it resulted in larvae with higher energy, protein, and lipid ratios but lowered carbohydrate, ash, and fiber.

Fourteen fatty acids were identified in the larvae, and the progressive inclusion of the grape residue resulted in significant increases in the contents of polyunsaturated fatty acids.

The replacement of the conventional diet with 25% grape residue is a viable alternative for larval mass gain, in contrast to the replacement with higher values. This substitution is equivalent to the conventional diet in many respects and improves several performance indices and larval nutrition. However, replacing the conventional diet with high contents of grape residue may reduce the benefits obtained. Substitutions with 50, 75, and 100% can be helpful if the goal is to increase potassium, phosphorus, PUFA, the n6/n3 ratio, and linoleic and linolenic acid in the larvae.

Insects are an increasingly common food, and the use of agro-industrial waste, such as grape residue in insect-rearing diets, is a sustainable approach with a potential reduction in environmental pollution, which contributes to the circular economy and the zero-waste program.

## Figures and Tables

**Figure 1 molecules-27-06963-f001:**
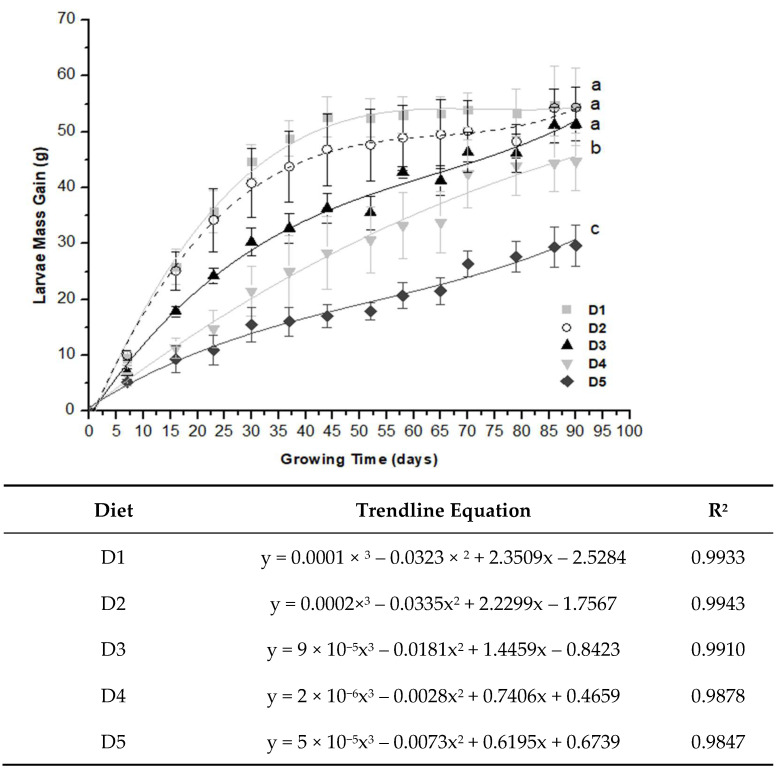
Cumulative mass gain of *Zophobas atratus* larvae fed a conventional diet with or without grape residue for 90 days, respective trend line equations, and regression determination coefficients (R^2^). Diets: D1 = 100% 70 g/100 g poultry feed and 30 g/100 g ground maize mixture (control); D2 = 75% D1 + 25% D5; D3 = 50% D1 + 50% D5; D4 = 25% D1 + 75% D5; D5 = 100% grape juice production residue. Different letters indicate significant differences by Tukey’s test (*p* ≤ 0.05; n = 3).

**Figure 2 molecules-27-06963-f002:**
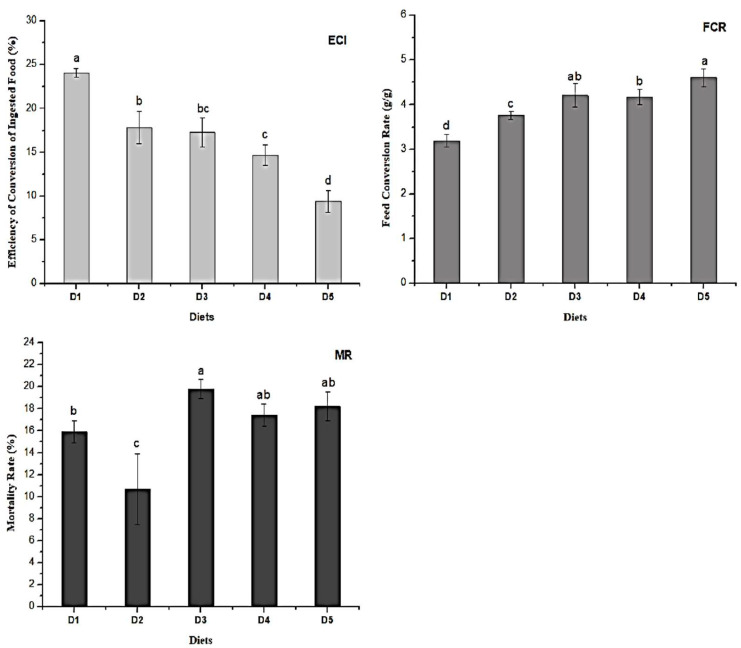
Efficiency of Conversion of Ingested food (ECI), Feed conversion rate (FCR), and Mortality rate (MR) of *Zophobas atratus* larvae fed a conventional diet with or without grape residue for 90 days. Diets: D1 = 100% 70 g/100 g poultry feed and 30 g/100 g ground maize mixture (control); D2 = 75% D1 + 25% D5; D3 = 50% D1 + 50% D5; D4 = 25% D1 + 75% D5; D5 = 100% grape juice production residue. Different letters indicate significant differences by Tukey’s test (*p* ≤ 0.05; n = 3).

**Figure 3 molecules-27-06963-f003:**
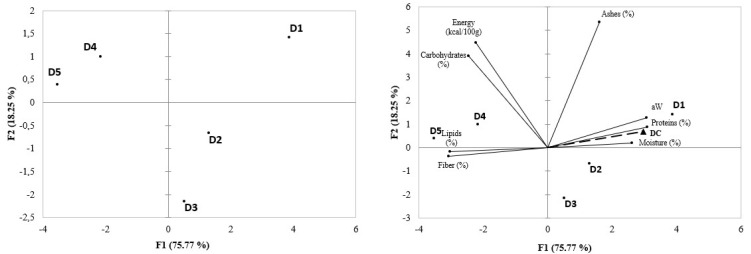
Analysis of main components of mass gain (DC) of *Zophobas atratus* larvae fed with a conventional diet with or without grape residue (D1 to D5) for 90 days and the parameters of the nutritional composition of the diets. Diets: D1 = 100% 70 g/100 g poultry feed and 30 g/100 g ground maize mixture (control); D2 = 75% D1 + 25% D5; D3 = 50% D1 + 50% D5; D4 = 25% D1 + 75% D5; D5 = 100% grape juice production residue.

**Table 1 molecules-27-06963-t001:** Macronutrient composition, water activity, and energy value of diets with and without grape residue for *Zophobas atratus* larvae growth.

Parameter * (g/100 g)	Diet **				
	D1 *	D2	D3	D4	D5
Moisture	8.46 ± 0.05 ^e^	9.74 ± 0.03 ^d^	11.02 ± 0.02 ^c^	12.31 ± 0.03 ^b^	13.59 ± 0.05 ^a^
Carbohydrate	73.24 ± 0.13 ^a^	69.50 ± 0.09 ^b^	65.77 ± 0.06 ^c^	62.03 ± 0.03 ^d^	58.30 ± 0.01 ^e^
Protein	10.66 ± 0.04 ^a^	8.96 ± 0.03 ^b^	7.06 ± 0.02 ^c^	5.56 ± 0.02 ^d^	3.86 ± 0.03 ^e^
Lipid	1.44 ± 0.02 ^e^	2.23 ± 0.02 ^d^	3.02 ± 0.03 ^c^	3.81 ± 0.04 ^b^	4.60 ± 0.06 ^a^
Ashes	3.72 ± 0.03 ^e^	3.83 ± 0.03 ^d^	3.95 ± 0.03 ^c^	4.06 ± 0.03 ^b^	4.17 ± 0.03 ^a^
Fiber	2.49 ± 0.01 ^e^	5.74 ± < 0.01 ^d^	8.99 ± < 0.01 ^c^	12.23 ± 0.01 ^b^	15.48 ± 0.02 ^a^
WA	0.710 ± < 0.01 ^a^	0.600 ± < 0.01 ^b^	0.560 ± 0.01 ^c^	0.520 ± < 0.01 ^d^	0.460 ± < 0.01 ^e^
Energy (kcal/100 g)	348.53 ± 0.23 ^a^	333.90 ± 0.15 ^b^	319.27 ± 0.26 ^c^	304.64 ± 0.43 ^d^	290.01 ± 0.61 ^e^

* Wet basis. ** Values expressed as arithmetic mean ± standard deviation (n = 3). WA: water activity. Diets: D1 = 100% 70 g/100 g poultry feed and 30 g/100 g ground maize mixture (control); D2 = 75% D1 + 25% D5; D3 = 50% D1 + 50% D5; D4 = 25% D1 + 75% D5; D5 = 100% grape juice production residue. Different letters in the row indicate significant differences by Tukey’s test (*p* ≤ 0.05; *n* = 3).

**Table 2 molecules-27-06963-t002:** Daily mass gain of *Zophobas atratus* larvae fed a conventional diet with or without grape residue from zero to 30, from 31 to 60, and from 61 to 90 days of growth.

Mass Gain (g/Day)	Time Course (Day)	Diet				
		D1	D2	D3	D4	D5
DMG	0 to 30	1.49 ± 0.10 ^aA^	1.36 ± 0.21 ^aA^	1.01 ± 0.08 ^bA^	0.72 ± 0.15 ^cA^	0.51 ± 0.10 ^dA^
DMG	31 to 60	0.14 ± 0.02 ^aB^	0.14 ± 0.03 ^aB^	0.18 ± 0.06 ^aB^	0.02 ± 0.04 ^cC^	0.09 ± 0.04 ^bB^
DMG	61 to 90	0.02 ± 0.06 ^aC^	0.06 ± 0.08 ^aB^	0.12 ± 0.02 ^aB^	0.13 ± 0.02 ^aB^	0.10 ± 0.04 ^aB^

DMG = daily mass gain of larvae divided by cultivation period. Diets: D1 = 100% 70 g/100 g poultry feed and 30 g/100 g ground maize mixture (control); D2 = 75% D1 + 25% D5; D3 = 50% D1 + 50% D5; D4 = 25% D1 + 75% D5; D5 = 100% grape juice production residue. Different lowercase letters in the row and uppercase letters in the column indicate significant differences by Tukey’s test (*p* ≤ 0.05; n = 3).

**Table 3 molecules-27-06963-t003:** Physicochemical characterization of *Zophobas atratus* larvae fed with a conventional diet with or without grape residue for 90 days.

Parameter ^1^ (g/100 g)	Diet				
	D1	D2	D3	D4	D5
Carbohydrate ^2^	3.58 ± 0.24 ^bc^	2.45 ± 0.57 ^c^	4.91 ± 0.38 ^b^	5.49 ± 1.26 ^ab^	7.37 ± 2.88 ^a^
Protein	41.90 ± 0.30 ^ab^	42.60 ± 0.62 ^a^	42.26 ± 0.40 ^a^	40.90 ± 1.24 ^ab^	39.58 ± 2.69 ^b^
Lipid	43.91 ± 0.37 ^b^	45.58 ± 0.03 ^a^	44.04 ± 0.05 ^b^	43.13 ± 0.04 ^c^	40.50 ± 0.05 ^d^
Ashes	3.02 ± 0.09 ^c^	3.04 ± 0.05 ^c^	3.06 ± 0.04 ^c^	3.43 ± 0.02 ^b^	3.93 ± 0.22 ^a^
Fiber	7.60 ± 0.28 ^b^	6.33 ± 0.04 ^d^	5.72 ± 0.04 ^e^	7.05 ± 0.03 ^c^	8.63 ± 0.04 ^a^
Water activity ^3^	0.060 ± 0.004 ^a^	0.047 ± 0.002 ^b^	0.063 ± 0.003 ^a^	0.052 ± 0.002 ^b^	0.046 ± 0.001 ^b^
Energy (kcal/100 g)	577.06 ± 1.98 ^c^	590.42 ± 0.23 ^a^	585.07 ± 0.37 ^b^	573.72 ± 0.10 ^d^	552.27 ± 0.87 ^e^

^1^ Dry basis. ^2^ Carbohydrate = value obtained by difference. Diets: D1 = 100% 70 g/100 g poultry feed and 30 g/100 g ground maize mixture (control); D2 = 75% D1 + 25% D5; D3 = 50% D1 + 50% D5; D4 = 25% D1 + 75% D5; D5 = 100% grape juice production residue. ^3^ Water activity from freeze-dried larvae. Values were expressed as arithmetic mean ± standard deviation (n = 3). Different letters in the row indicate significant differences by Tukey’s test (*p* ≤ 0.05; n = 3).

**Table 4 molecules-27-06963-t004:** Fatty acid profile of *Zophobas atratus* larvae fed with a conventional diet with and without grape residue (D1 to D5) for 90 days (dry basis).

Fatty Acids (%)		Diet ^1^				
		D1	D2	D3	D4	D5
Saturated	C8:0	0.36 ± 0.02 ^b^	0.37 ± 0.01 ^b^	0.48 ± 0.05 ^a^	0.48 ± 0.01 ^a^	0.46 ± 0.02 ^a^
	C10:0	0.07 ± 0.01 ^a^	0.10 ± 0.02 ^a^	0.09 ± 0.01 ^a^	0.09 ± 0.01 ^a^	0.09 ± 0.01 ^a^
	C11:0	0.15 ± 0.01 ^b^	0.17 ± 0.02 ^ab^	0.19 ± 0.01 ^a^	0.17 ± 0.01 ^ab^	0.17 ± 0.01 ^ab^
	C12:0	0.08 ± 0.01 ^a^	0.08 ± 0.01 ^a^	0.07 ± 0.01 ^ab^	0.04 ± 0.01 ^b^	0.05 ± 0.01 ^b^
	C14:0	1.18 ± 0.05 ^a^	1.18 ± 0.03 ^a^	1.14 ± 0.04 ^a^	0.78 ± 0.02 ^c^	1.02 ± 0.01 ^b^
	C15:0	0.15 ± 0.01 ^c^	0.18 ± 0.01 ^b^	0.16 ± 0.01 ^bc^	0.17 ± 0.01 ^b^	0.22 ± 0.01 ^a^
	C16:0	34.37 ± 0.36 ^a^	32.91 ± 0.27 ^b^	32.40 ± 0.26 ^b^	29.90 ± 0.27 ^c^	28.22 ± 0.09 ^d^
	C17:0	0.34 ± 0.04 ^c^	0.41 ± 0.03 ^bc^	0.40 ± 0.03 ^bc^	0.45 ± 0.03 ^b^	0.56 ± 0.01 ^a^
	C18:0	6.59 ± 0.17 ^c^	7.40 ± 0.12 ^c^	7.15 ± 0.74 ^c^	8.38 ± 0.25 ^b^	9.59 ±0.06 ^a^
Monounsaturated	C16:1n7	0.85 ± 0.03 ^a^	0.82 ± 0.03 ^a^	0.72 ± 0.04 ^ab^	0.54 ± 0.03 ^b^	0.60 ± 0.16 ^b^
	C17:1n7	0.14 ± 0.04 ^a^	0.14 ± 0.03 ^a^	0.12 ± 0.01 ^ab^	0.07 ± 0.01 ^b^	0.10 ± 0.02 ^ab^
	C18:1n-9	33.81 ± 0.25 ^a^	32.42 ± 0.62 ^a^	32.47 ± 0.51 ^a^	29.19 ± 1.05 ^b^	27.89 ± 0.42 ^b^
Polyunsaturated	C18:2n-6	19.40 ± 0.13 ^c^	21.12 ± 0.15 ^b^	21.78 ± 1.25 ^b^	26.83 ± 0.54 ^a^	27.79 ± 0.16 ^a^
	C18:3n-3	1.40 ± 0.01 ^c^	1.71 ± 0.07 ^b^	1.78 ± 0.15 ^b^	1.91 ± 0.04 ^b^	2.17 ± 0.08 ^a^
	∑ SFA	43.28 ± 0.42 ^a^	42.80 ± 0.19 ^a^	42.08 ± 0.84 ^a^	40.46 ± 0.50 ^b^	40.38 ± 0.08 ^b^
	∑ MUFA	34.80 ± 0.25 ^a^	33.38 ± 0.63 ^a^	33.31 ± 0.53 ^a^	29.80 ± 1.05 ^b^	28.59 ± 0.26 ^b^
	∑ PUFA	20.80 ± 0.12 ^c^	22.82 ± 0.09 ^b^	23.56 ± 1.38 ^b^	28.74 ± 0.50 ^a^	29.96 ± 0.16 ^a^
	n6/n3	13.88 ± 0.18 ^a^	12.45 ± 0.63 ^b^	12.28 ± 0.47 ^b^	14.08 ± 0.62 ^a^	12.89 ± 0.47 ^ab^
	Other Fatty Acids	1.15 ± 0.15 ^a^	1.04 ± 0.40 ^a^	1.08 ± 0.10 ^a^	1.02 ± 0.10 ^a^	1.16 ± 0.03 ^a^
	IA	0.76 ± 0.03 ^a^	0.74 ± 0.02 ^a^	0.71 ± < 0.01 ^b^	0.63 ± 0.02 ^d^	0.67 ± 0.01 ^c^
	IT	1.58 ± 0.06 ^a^	1.59 ± 0.04 ^a^	1.53 ± 0.03 ^a^	1.42 ± 0.03 ^b^	1.60 ± 0.04 ^a^

SFA = saturated fatty acids; MUFA = monounsaturated fatty acids; PUFA = polyunsaturated fatty acids; IA = index of atherogenicity; IT = index of thrombogenicity. ^1^ Diet: D1 = 100% 70 g/100 g poultry feed and 30 g/100 g ground maize mixture (control); D2 = 75% D1 + 25% D5; D3 = 50% D1 + 50% D5; D4 = 25% D1 + 75% D5; D5 = 100% grape juice production residue. Values were expressed as arithmetic mean ± standard deviation (n = 3). Different letters in the row indicate significant differences by Tukey’s test (*p* ≤ 0,05; n = 3). The fatty acid profile of larvae fed with the minor addition of grape residues (control, D2, and D3) was higher for SFA and MUFA (*p* ≤ 0.05). In contrast, when fed with higher addition of grape residues (D4 and D5), it was higher for PUFA and n6/(*p* ≤ 0.05). The other fatty acids were similar for all treatments and control (Table 4). The IA value was higher in D2 and control than the other treatments (*p* ≤ 0.05) and lower in treatments with higher addition of grape residue in the diet, such as D3, D4, and D5 (*p* ≤ 0.05). IT value was the same for all treatments and control, except for D4 (Table 4). The oleic acid contents in the present study ranged from 28 to 34%. The values obtained were similar to the results of Zhang et al. [25], from 21 to 34%, but lower than those reported by Ruschioni et al. [28], from 45 to 58%, and by van Broekhoven et al. [20] from 40 to 58% oleic acid content. The greater addition of grape residue in the diet of the larvae resulted in a lower amount of oleic acid in the larvae. However, the progressive increase in residue in the diets resulted in a significant increase in the contents of essential fatty acids C18:2n6 (43.3%) and C18:3n3 (55%) in D5 when compared to the control.

**Table 5 molecules-27-06963-t005:** Metal and non-metal composition in *Zophobas atratus* larvae fed a conventional diet with or without grape residue (D1 to D5) for 90 days (dry basis).

Metal and Non-Metal (mg/100 g)	Diet				
	D1	D2	D3	D4	D5
Calcium	49.69 ± 0.40 ^a^	49.36 ± 0.07 ^ab^	48.23 ± 0.06 ^bc^	47.52 ± 0.06 ^cd^	46.78 ± 0.94 ^d^
Potassium	1058.83 ± 7.88 ^e^	1125.45 ± 4.54 ^d^	1354.59 ± 4.96 ^c^	1495.82 ± 7.44 ^b^	1611.40 ± 2.76 ^a^
Phosphor	555.3 ± 5.51 ^e^	607.94 ± 2.99 ^d^	648.84 ± 7.29 ^c^	701.41 ± 2.63 ^b^	731.51 ± 2.34 ^a^
Magnesium	133.57 ± 3.27 ^c^	144.77 ± 1.96 ^b^	146.78 ± 0.57 ^b^	160.04 ± 0.82 ^a^	161.71 ± 0.85 ^a^
Manganese	0.9 ± 0.01 ^a^	0.90 ± 0.01 ^a^	0.91 ± 0.02 ^a^	0.91 ± 0.01 ^a^	0.91 ± 0.01 ^a^
Iron	15.74 ± 0.23 ^a^	15.71 ± 0.21 ^a^	15.69 ± 0.04 ^a^	15.70 ± 0.02 ^a^	15.68 ± 0.02 ^a^
Selenium *	<2.00	<2.00	<2.00	<2.00	<2.00
Sodium	2545.96 ± 4.77 ^a^	2512.86 ± 8.50 ^b^	2412.76 ± 14.17 ^c^	2352.47 ± 8.50 ^d^	2301.85 ± 18.21 ^e^
Zinc	7.14 ± 0.11 ^a^	6.92 ± 0.08 ^b^	6.95 ± 0.05 ^b^	6.85 ± 0.02 ^bc^	6.72 ± 0.03 ^c^
Arsenic *	<1.50	<1.50	<1.50	<1.50	<1.50
Cadmium *	<1.00	<1.00	<1.00	<1.00	<1.00
Cobalt *	<1.50	<1.50	<1.560	<1.50	<1.50
Copper *	<1.50	<1.50	<1.50	<1.50	<1.50
Nickel *	<2.00	<2.00	<2.00	<2.00	<2.00

Diets: D1 = 100% 70 g/100 g poultry feed and 30 g/100 g ground maize mixture (control); D2 = 75% D1 + 25% D5; D3 = 50% D1 + 50% D5; D4 = 25% D1 + 75% D5; D5 = 100% grape juice production residue. Values were expressed as arithmetic mean ± standard deviation (n = 3). Different letters in the row indicate significant differences by Tukey’s test (*p* ≤ 0.05; n = 3). * Element detection limits.

**Table 6 molecules-27-06963-t006:** Diet formulation and coding for feeding *Zophobas atratus* larvae.

Feedstock	Diet (%; Mass/Mass)				
	D1 *	D2	D3	D4	D5
Grape juice production residue	0	25	50	75	100
Mixture of 70 g/100 g commercial poultry feed ** and 30 g/100 g commercial ground maize grain	100	75	50	25	0

* Control; ** Poultry feed = commercial poultry feed for egg-laying chicken.

## Data Availability

Not applicable.

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
