# Peer review of "Utilization of Agro-Industrial Residues in the Rearing and Nutritional Enrichment of Zophobas atratus Larvae: New Food Raw Materials"

_molecules, 2022, doi:10.3390/molecules27206963_

Round 1

Reviewer 1 Report

Nascimento et al. aimed to detect an effect of replacement of conventional diet with partial grape residue on larval growth and development, and other larvae performance indices and nutritional values of Zophobas morio. It is an interesting work, which is not only good for the related researches in insect but also for the edible insect industry. Their conclusions are consistent with the evidences presented.

Comments:

1.       Why the mortality of D2 in Figure 2 was lowest but not D1? It seems that the composite indicator of D1 was the best among the larvae feeding with different diets.

2.       In table 2, the lipid contents in the grape-residue supplied larvae were gradually increased, but why the energy was decreased so much?

3.        I wonder how whether feeding on grape residues will prolong the growth period of larvae? From Figure 1, it seems to take much longer time for the replaced diet groups to reach to the highest larval mass gain.

Author Response

I am attaching the revised version of the manuscript. We appreciate the reviewers' valuable contributions to the manuscript and would like to inform you that all suggestions have been accepted and changed in the manuscript. In addition, we have revised the English used all over the manuscript. Thank you for the opportunity to disseminate our findings in this prestigious journal, and please see below for our responses to reviewer 1.

Answer to Reviewer 1:

Reviewer question/comment 1:

Comments and Suggestions for Authors

Nascimento et al. aimed to detect an effect of replacement of conventional diet with partial grape residue on larval growth and development and other larvae performance indices and nutritional values of Zophobas morio. It is an interesting work, which is not only good for the related researches in insects but also for the edible insect industry. Their conclusions are consistent with the evidences presented.”

Answer:

Thank you for your consideration. It was nice to receive your comments on the manuscript. 

“Comments:

  1. Why the mortality of D2 in Figure 2 was lowest but not D1? It seems that the composite indicator of D1 was the best among the larvae feeding with different diets.”

Answer:

Among the diets offered to the larvae, the lowest mortality was observed in D2 (25% residue). This may have occurred because the composition of this diet resulted in a proportion of compounds favorable and sufficient for larval growth, being able to control natural competitors, including antagonistic microorganisms, thus promoting a more favorable breeding environment compared to D1 (control).

“2.       In table 2, the lipid contents in the grape-residue supplied larvae were gradually increased, but why the energy was decreased so much?”

Answer:

The energy values in the diets were reduced because, at the same time, the lipid content gradually increased with the inclusion of grape residue, the carbohydrate and protein contents were reduced. In the methodology proposed by Atwater & Woods (1896) for energy calculation, carbohydrates and proteins are multiplied by 4 kcal/g and lipids by 9 kcal/g, respectively. Thus, for diets D2, D3, D4, and D5, summing only protein and carbohydrates, parameters that gradually decreased, we observed a reduction in the energy content (313.84; 291.32; 270.36; 248.64), directly affecting the final values.

“3.        I wonder how whether feeding on grape residues will prolong the growth period of larvae? From Figure 1, it seems to take much longer time for the replaced diet groups to reach to the highest larval mass gain.”

Answer:

The addition of grape residue in the diet reduced the mass gain of the larvae over the feeding time (Figure 1). By extrapolation of the mathematical equation, larvae fed, for example, 100% grape residue (D5) would require an additional 30 days of feeding to achieve the greatest mass gains obtained with the 25% grape residue replacement in the D2 diet.

Reviewer 2 Report

Authors wored on a very impotant nowadays problem - how to provde an alternative food source for increasing number of people.

The main aim of that study was to evaluate the performance and nutritional characterization of Zophobas atratus larvae fed with different proportions of grape residue.

The manuscript has been preaperd very accurately, both in terms of editorial and scientific content. Methods have been precisley described and presented in a form of logically divided parts what helped with their analyses.

The authors properly came up with the right conclusion that only the  replacement of 25% of the conventional diet by grape residue showed similar results to the control and was more effective than the replacements of 50%, 75% and 100%.

Thus there are only a few minor mistakes which have been made by Authors - all within editorial category. 

Authors should standardize the citation of the bibliography - once they list all authors and other times they use the abbreviation et al.

In lines  - should be 

55 - BednáÅ™ová et al.; Kim et al.

57 - Nowk et al.

60 - van der Spiegel et al.

63 - Payne et al.

67 - van Broekhoven et al.

78 - Oonicx et al.

161 - Jaansen et al.

268 - is Chapman et al., 1998; Li, Zhao, & Liu should be Chapman, 1998; Li et al.

352 - is Herrero & Thornton, et al2013 should be Herrero & Thornton, 2013

409 - is Makkar, 409 Tran, Heuzé, & Ankers (2014) should be Makkar et al.

414-415 is Klasing, Thacker, Lopez, & 414 Calvert should be Klasing et al.

446 - is Ravzanaadii, Kim, Choi, Hong, & Kim, should be Ravzanaadii et al.

447 - is Yoo, Hwang, Goo, & Yun, 2013). According to Gao, Li, Zhang, & Hao should be Yoo et al., 2013). According to Gao et al. 

461 - is Tzompa-Sosa, Yi, van Valenberg, van Boekel, & Lakemond, 2014 should be Tzompa-Sosa et al., 2014

523 - is Van Broekhoven should be van Broekhoven

529-530 is  Raksakantong, Meeso, Kubola, & Siriamornpun, 2010; ZieliÅ„ska, Baraniak, KaraÅ›, 529 RybczyÅ„ska, & Jakubczyk et al., 2015 should be Raksakantong et al., 2010; ZieliÅ„ska et al., 2015 

551-552 - is Ramos-Bueno, González-Fernández, Sánchez-Muros-Lozano, 551 García-Barroso, & Guil-Guerrero (2016)should be Ramos-Bueno et al.,  (2016)

Moreover:

line 65 - is Coleoptera: Tenebrionidae should be Coleoptera: Tenebrionidae

line 135 - is Solanum tuberosum should be Solanum tuberosum

line 410-411 is Tenebrionidae should be Tenebrionidae

line 498 and 538 is Coleoptera should be Coleoptera

A finally reference No. 38 (lines 699-700) should be cited after Ulbricht (ref. No. 56) 

Author Response

Please find attached the revised version of the manuscript. We appreciate the reviewers' valuable contributions to the manuscript and would like to inform you that all suggestions have been accepted and changed in the manuscript. In addition, we have highlighted in red all the changes and revised the English used all over the manuscript. Thank you for the opportunity to disseminate our findings in this prestigious journal, and please see below for our responses to reviewer 2.

Answer to Reviewer 2:

Reviewer question/comment 1:

“Authors wored on a very impotant nowadays problem - how to provde an alternative food source for increasing number of people.

The main aim of that study was to evaluate the performance and nutritional characterization of Zophobas atratus larvae fed with different proportions of grape residue.

The manuscript has been preaperd very accurately, both in terms of editorial and scientific content. Methods have been precisley described and presented in a form of logically divided parts what helped with their analyses.

The authors properly came up with the right conclusion that only the  replacement of 25% of the conventional diet by grape residue showed similar results to the control and was more effective than the replacements of 50%, 75% and 100%.

Thus there are only a few minor mistakes which have been made by Authors - all within editorial category.”

Answer:

Thank you for your consideration. It was nice to receive your comments on the manuscript. 

Reviewer question/comment 2:

“Comments:

Authors should standardize the citation of the bibliography - once they list all authors and other times they use the abbreviation et al.

In lines  - should be 

55 - BednáÅ™ová et al.; Kim et al.

57 - Nowk et al.

60 - van der Spiegel et al.

63 - Payne et al.

67 - van Broekhoven et al.

78 - Oonicx et al.

161 - Jaansen et al.

268 - is Chapman et al., 1998; Li, Zhao, & Liu should be Chapman, 1998; Li et al.

352 - is Herrero & Thornton, et al2013 should be Herrero & Thornton, 2013

409 - is Makkar, 409 Tran, Heuzé, & Ankers (2014) should be Makkar et al.

414-415 is Klasing, Thacker, Lopez, & 414 Calvert should be Klasing et al.

446 - is Ravzanaadii, Kim, Choi, Hong, & Kim, should be Ravzanaadii et al.

447 - is Yoo, Hwang, Goo, & Yun, 2013). According to Gao, Li, Zhang, & Hao should be Yoo et al., 2013). According to Gao et al. 

461 - is Tzompa-Sosa, Yi, van Valenberg, van Boekel, & Lakemond, 2014 should be Tzompa-Sosa et al., 2014

523 - is Van Broekhoven should be van Broekhoven

529-530 is  Raksakantong, Meeso, Kubola, & Siriamornpun, 2010; ZieliÅ„ska, Baraniak, KaraÅ›, 529 RybczyÅ„ska, & Jakubczyk et al., 2015 should be Raksakantong et al., 2010; ZieliÅ„ska et al., 2015 

551-552 - is Ramos-Bueno, González-Fernández, Sánchez-Muros-Lozano, 551 García-Barroso, & Guil-Guerrero (2016)should be Ramos-Bueno et al.,  (2016)”

Answer:

We standardized the citation as suggested.

Reviewer question/comment 3:

“Moreover:

line 65 - is Coleoptera: Tenebrionidae should be Coleoptera: Tenebrionidae

line 135 - is Solanum tuberosum should be Solanum tuberosum

line 410-411 is Tenebrionidae should be Tenebrionidae

line 498 and 538 is Coleoptera should be Coleoptera

A finally reference No. 38 (lines 699-700) should be cited after Ulbricht (ref. No. 56)”

Answer:

We have made all the changes as suggested.

Thank you for your consideration and suggestions.

Reviewer 3 Report

I am very grateful you for the invitation to review the manuscript molecules-1939521 by Nascimento and coauthors "Utilization of agro-industrial residues in the rearing and nutritional enrichment of Zophobas atratus larvae: new food raw materials”. The study evaluated the performance and nutritional characterization of Zophobas atratus larvae fed with different proportions of grape residue (25, 50, 75, and 100 wt %) in the conventional diet (control). The work is interesting but needs adjustments to increase the quality of the material.

Comments:

- Title: Adjust the title to the journal’s format style (differentiate between uppercase and lowercase).

- Line 45: Review the citation format of references.

- The citation format throughout the text is inappropriate. Verify the “Instructions for Authors”.

- The sequence of items disagrees with the journal's authors’ guide.

- Abstract, Line 26: Change “alternative” to “potential alternative”.

- Abstract: Briefly include the steps for identifying the compounds (obtaining, extracting and other steps).

- Introduction, Line 45: The proposal to use insects as an alternative source of nutrients is mentioned ten years ago. Authors must present a current scenario.

- Introduction: Even highlighting the advantages, the authors should cite problems related to insect production as an alternative (no process is 100% problem-free).

- Introduction: Authors should highlight ideal growth conditions for the insect in question.

- 2.1. Biological material: It is not clear whether the poultry feed cited is used as a replacement matrix or used at a pre-analysis time. In the next item, a new feed formulation is presented. Properly specify the application of each one.

- Material: Authors must present the composition of the residue from the production of grape juice to provide information on the influence of the treatment.

- MM: Standardize the use of terms “Proximate composition” and “Physicochemical analysis”.

- Please unify Table 1 and 2 to facilitate comparison.

- Results and Discussion: Authors should include discussions of growth metabolism or associated factors as it is reported and compared to other sources.

- Results and Discussion: Include information about how each dietary component influences growth.

- Results and Discussion: It is unclear how much waste is converted and whether the alternative insect production system achieves conversion efficiency as highlighted in the introduction.

- Results and Discussion, Line 330: Indicate possible active components since phenolic compounds were reported as a negative factor for insect production.

- Line 395: Review de sentence “energy (p ≤ 0.05). 05),”.

- Line 432: Specify whether water activity values refer to freeze-dried larvae, as values are extremely low.

- Results and Discussion: Please include some information on acceptance of alternative insect consumption and future perspectives.

Author Response

Please find attached the revised version of the manuscript. We appreciate the reviewers' valuable contributions to the manuscript and would like to inform you that all suggestions have been accepted and changed in the manuscript. In addition, we have highlighted in red all the changes and revised the English used all over the manuscript. Thank you for the opportunity to disseminate our findings in this prestigious journal, and please see below for our responses to reviewer 3.

Answer to Reviewer 3:

Reviewer question/comment 1:

“Comments and Suggestions for Authors

I am very grateful you for the invitation to review the manuscript molecules-1939521 by Nascimento and coauthors "Utilization of agro-industrial residues in the rearing and nutritional enrichment of Zophobas atratus larvae: new food raw materials”. The study evaluated the performance and nutritional characterization of Zophobas atratus larvae fed with different proportions of grape residue (25, 50, 75, and 100 wt %) in the conventional diet (control). The work is interesting but needs adjustments to increase the quality of the material.”

Answer:

Thank you for your consideration. It was nice to receive your comments on the manuscript. 

Reviewer question/comment 2:

“Comments:

- Title: Adjust the title to the journal’s format style (differentiate between uppercase and lowercase).”

Answer:

We have changed as suggested.

Reviewer question/comment 3:

“- Line 45: Review the citation format of references.

- The citation format throughout the text is inappropriate. Verify the “Instructions for Authors”.”

Answer:

We standardized the citations and references.

Reviewer question/comment 4:

“- The sequence of items disagrees with the journal's authors’ guide.”

Answer:

We have changed as suggested.

Reviewer question/comment 5:

“- Abstract, Line 26: Change “alternative” to “potential alternative”.”

Answer:

We have changed as suggested.

Reviewer question/comment 6:

“- Abstract: Briefly include the steps for identifying the compounds (obtaining, extracting and other steps).”

Answer:

We have added information as suggested and rewrote the text to still maintain 200 words.

Reviewer question/comment 7:

“- Introduction, Line 45: The proposal to use insects as an alternative source of nutrients is mentioned ten years ago. Authors must present a current scenario.”

Answer:

We have added information as suggested.

Reviewer question/comment 8:

“- Introduction: Even highlighting the advantages, the authors should cite problems related to insect production as an alternative (no process is 100% problem-free).”

Answer:

We have added information as suggested.

Reviewer question/comment 9:

“- Introduction: Authors should highlight ideal growth conditions for the insect in question.”

Answer:

We have added information as suggested.

Reviewer question/comment 10:

“- 2.1. Biological material: It is not clear whether the poultry feed cited is used as a replacement matrix or used at a pre-analysis time. In the next item, a new feed formulation is presented. Properly specify the application of each one.”

Answer:

We have changed it for a better understanding of the text.

Reviewer question/comment 11:

“- Material: Authors must present the composition of the residue from the production of grape juice to provide information on the influence of the treatment.”

Answer:

We have changed the text for a better understanding of this component. However, treatment D5 which is 100% grape juice residue was analyzed, and the results were presented in the tables.

Reviewer question/comment 12:

“- MM: Standardize the use of terms “Proximate composition” and “Physicochemical analysis”.”

Answer:

We have changed as suggested.

Reviewer question/comment 13:

“- Please unify Table 1 and 2 to facilitate comparison.”

Answer:

We analyzed the viability of unifying the tables but because they are in different sectors of the manuscript and the unification would result in a complex table, making it difficult to visualize the data, we have kept the current form.

Reviewer question/comment 14:

“- Results and Discussion: Authors should include discussions of growth metabolism or associated factors as it is reported and compared to other sources.”

Answer:

We have added it as suggested.

Reviewer question/comment 15:

“- Results and Discussion: Include information about how each dietary component influences growth.”

Answer:

Our main objective was to evaluate the effect of replacing the conventional diet with grape residue, and an evaluation of each component of the diet would require further research. However, speculations on the effect of the components of the grape residue have been added.

Reviewer question/comment 16:

“- Results and Discussion: It is unclear how much waste is converted and whether the alternative insect production system achieves conversion efficiency as highlighted in the introduction.”

Answer:

We have added it as suggested.

Reviewer question/comment 17:

“- Results and Discussion, Line 330: Indicate possible active components since phenolic compounds were reported as a negative factor for insect production.”

Answer:

We have added it as suggested.

Reviewer question/comment 18:

“- Line 395: Review de sentence “energy (p ≤ 0.05). 05),”.”

Answer:

We have changed as suggested.

Reviewer question/comment 19:

“- Line 432: Specify whether water activity values refer to freeze-dried larvae, as values are extremely low.”

Answer:

We have added the information for a better understanding of the text.

Reviewer question/comment 20:

“- Results and Discussion: Please include some information on acceptance of alternative insect consumption and future perspectives.”

Answer:

We have added it as suggested.

Round 2

Reviewer 3 Report

After carefully checking the responses and the revisions, the manuscript is suggested to be accepted by Molecules;